# DL2: Training and Querying Neural Networks with Logic

## Abstract

We present DL2, a system for training and querying neural networks with logical constraints. DL2 is more expressive than prior work and can capture a richer class of constraints on inputs, outputs and internals of models. Using DL2, one can declaratively specify domain knowledge to be enforced during training or pose queries on the model with the goal of finding inputs that satisfy a given constraint. DL2 works by translating logical constraints into a differentiable loss with desirable mathematical properties, then minimized with standard gradient-based methods. Our evaluation demonstrates that DL2 can express interesting constraints beyond the reach of prior work, leading to improved prediction accuracy.

## 1 Introduction

With the success of neural networks across a wide range of important application domains, a key challenge that has emerged is that of making neural networks more reliable. Promising directions to address this challenge are incorporating constraints during training (Madry et al., 2017; Minervini et al., 2017) and inspecting already trained networks by posing specific queries (Goodfellow et al., 2014b; Pei et al., 2017; Xu et al., 2018)). While useful, these approaches are described and hardcoded to particular kinds of constraints, making their application to other settings difficult.

Inspired by prior work (e.g., Cohen et al. (2017); Fu & Su (2016); Hu et al. (2016); Bach et al. (2017)), we introduce a new method and system, called DL2 (acronym for Deep Learning with Differentiable Logic), which can be used to: (i) query networks for inputs meeting constraints, and (ii) train networks to meet logical specifications, all in a *declarative* fashion. Our constraint language can express rich combinations of arithmetic comparisons over inputs, neurons and outputs of neural networks using negations, conjunctions, and disjunctions. Thanks to its expressiveness, DL2 enables users to enforce domain knowledge during training or interact with the network in order to learn about its behavior via querying.

DL2 works by translating logical constraints into non-negative loss functions with two key properties: (P1) a value where the loss is zero is guaranteed to satisfy the constraints, and (P2) the resulting loss is differentiable almost everywhere. Combined, these properties enable us to solve the problem of querying or training with constraints by minimizing a loss with off-the-shelf optimizers.

**Training with DL2** To make optimization tractable, we exclude constraints on inputs that capture convex sets and include them as constraints to the optimization goal. We then optimize with projected gradient descent (PGD), shown successful for training with robustness constraints (Madry et al., 2017). The expressiveness of DL2 along with tractable optimization through PGD enables us to train with new, interesting constraints. For example, we can express constraints over probabilities which are *not* explicitly computed by the network. Consider the following:

$$\forall \boldsymbol{x}.\ p_{people}^{\theta}(\boldsymbol{x}) < \epsilon \vee p_{people}^{\theta}(\boldsymbol{x}) > 1 - \epsilon$$

This constraint, in the context of CIFAR-100, says that for any network input $\boldsymbol{x}$ (network is parameterized by $\theta$), the probability of *people* ($p_{people}$) is either very small or very large. However, CIFAR-100 does not have the class *people*, and thus we define it as a *function of other probabilities*, in particular: $p_{people} = p_{baby} + p_{boy} + p_{girl} + p_{man} + p_{woman}$. We show that with a similar constraint (but with 20 classes), DL2 increases the prediction accuracy of CIFAR-100 networks in the semi-supervised setting, outperforming prior work whose expressiveness is more restricted.

DL2 can capture constraints arising in both, classification and regression tasks. For example, GalaxyGAN (Schawinski et al., 2017), a generator of galaxy images, requires the network to respect constraints imposed by the underlying physical systems, e.g., flux: the sum of input pixels should equal the sum of output pixels. Instead of hardcoding such a constraint into the network in an ad hoc way, with DL2, this can now be expressed declaratively: $sum(\boldsymbol{x}) = sum(\text{GalaxyGAN}(\boldsymbol{x}))$.

**Global training** A prominent feature of DL2 is its ability to train with constraints that place restrictions on inputs *outside the training set*. Prior work on training with constraints (e.g., Xu et al. (2018)) focus on the given training set to *locally train* the network to meet the constraints. With DL2, we can, for the first time, query for inputs which are *outside* the training set, and use them to *globally train* the network. Previous methods that trained on examples outside the training set were either tailored to a specific task (Madry et al., 2017) or types of networks (Minervini et al., 2017). Our approach splits the task of global training between: (i) the optimizer, which trains the network to meet the constraints for the given inputs, and (ii) the oracle, which provides the optimizer with new inputs that aim to violate the constraints. To illustrate, consider the following Lipshcitz condition:

$$\forall \boldsymbol{z}^1 \in L_\infty(\boldsymbol{x}^1, \epsilon), \boldsymbol{z}^2 \in L_\infty(\boldsymbol{x}^2, \epsilon).||p^\theta(\boldsymbol{z}^1) - p^\theta(\boldsymbol{z}^2)||_2 < L||\boldsymbol{z}^1 - \boldsymbol{z}^2||_2$$

Here, for two inputs from the training set $(\boldsymbol{x}^1, \boldsymbol{x}^2)$, any point in their $\epsilon$-neighborhood $(\boldsymbol{z}^1, \boldsymbol{z}^2)$ must satisfy the condition. This constraint is inspired by recent works (e.g., Gouk et al. (2018); Balan et al. (2017)) which showed that neural networks are more stable if satisfying the Lipschitz condition.

**Querying with DL2** We also designed an SQL-like language which enables users to interact with the model by posing declarative queries. For example, consider the scenarios studied by a recent work (Song et al., 2018) where authors show how to generate adversarial examples with AC-GANs (Odena et al., 2016). The generator is used to create images from a certain class (e.g., 1) which fools a classifier (to classify as, e.g., 7). With DL2, this can be phrased as:

```
find n[100]
where n in [-1, 1],
      class(M_NN1(M_ACGAN_G(n, 1))) = 7
return M_ACGAN_G(n, 1)
```

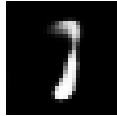

This query aims to find an input n $\in \mathbb{R}^{100}$ to the generator satisfying two constraints: its entries are between $-1$ and $1$ (enforcing a domain constraint) and it results in the generator producing an image, which it believes to be classified as 1 (enforced by `M_ACGAN_G(n, 1)`) but is classified by the network (`M_NN1`) as 7. DL2 automatically translates this query to a DL2 loss and optimizes it with an off-the-shelf optimizer (L-BFGS-B) to find solutions, in this case, the image to the right. Our language can naturally capture many prior works at the declarative level, including finding neurons responsible for a given prediction (Olah et al., 2018), inputs that differentiate two networks (Pei et al., 2017), and adversarial example generation (e.g., Szegedy et al. (2013)).

**Main Contributions** The DL2 system is based on the following contributions:

- An approach for training and querying neural networks with logical constraints based on translating these into a differentiable loss with desirable properties ((P1) and (P2)).
- A training procedure which extracts constraints on inputs that capture convex sets and includes them as PGD constraints, making optimization tractable.
- A declarative language for posing queries over neural network's inputs, outputs, and internal neurons. Queries are compiled into a differentiable loss and optimized with L-BFGS-B.
- An extensive evaluation demonstrating the effectiveness of DL2 in querying and training neural networks. Among other experimental results, we show for the first time, the ability to successfully train networks with constraints on inputs not in the training set.

## 2 RELATED WORK

Adversarial example generation (Pei et al., 2017; Goodfellow et al., 2014b) can be seen as a fixed query to the network, while adversarial training (Madry et al., 2017) aims to enforce a specific constraint. Most works aiming to train networks with logic impose soft constraints, often through an additional loss (Pathak et al., 2015; Xu et al., 2018); (Márquez-Neila et al., 2017) shows that hard constraints have no empirical advantage over soft constraints. Probabilistic Soft Logic (PSL) (Kimmig et al., 2012) translates logic into continuous functions over $[0, 1]$. As we show, PSL is not

amenable to gradient-based optimization as gradients may easily become zero. Hu et al. (2016) builds on PSL and presents a teacher-student framework which distills rules into the training phase. The idea is to formulate rule satisfaction as a convex problem with a closed-form solution. However, this formulation is restricted to rules over random variables and cannot express rules over probability distributions. In contrast, DL2 can express such constraints, e.g., $p_1 > p_2$, which requires the network probability for class 1 is greater than for 2. Also, the convexity and the closed-form solution stem from the linearity of the rules in the network's output, meaning that non-linear constraints (e.g., Lipschitz condition, expressible with DL2) are fundamentally beyond the reach of this method. The work of Xu et al. (2018) is also restricted to constraints over random variables and is intractable for complicated constraints. Fu & Su (2016) reduces the satisfiability of floating-point formulas into numerical optimization, however, their loss is not differentiable and they do not support constraints on distributions. Finally, unlike DL2, no prior work supports constraints for regression tasks.

## 3 FROM LOGIC TO A DIFFERENTIABLE LOSS

We now present our constraint language and show how to translate constraints into a differentiable loss. To simplify presentation, we treat all tensors as vectors with matching dimensions.

**Logical Language** Our language consists of quantifier-free constraints which can be formed with conjunction ($\wedge$), disjunction ($\vee$) and negation ($\neg$). Atomic constraints (literals) are comparisons $\bowtie$ of terms (here $\bowtie \in \{=, \neq, \leq, <, \geq, >\}$). Comparisons are defined for scalars and applied element-wise on vectors. A term $t$ is: (i) A variable $z$ or a constant $c$, representing real-valued vectors; constants can be samples from the dataset. (ii) An expression over terms, including arithmetic expressions or function applications $f \colon \mathbb{R}^m \to \mathbb{R}^n$, for $m, n \in \mathbb{Z}^+$. Functions can be defined overvariables, constants, and network parameters $\theta^1, \ldots, \theta^l$. Functions can be the application of a network with parameters $\theta$, the application of a specific neuron, or a computation over multiple networks. The only assumption on functions is that they are differentiable (almost everywhere) in the variables and network parameters. We write $t(z^1, \ldots, z^k, c^1, \ldots, c^j, \theta^1, \ldots \theta^l)$ to emphasize the variables, constants, and network parameters that $t$ can be defined over (that is, $t$ may refer to only a subset of these symbols). We sometimes omit the constants and network parameters (which are also constant) and abbreviate variables by $\bar{z}$, i.e., we write $t(\bar{z})$. Similarly, we write $\varphi(\bar{z})$ to denote a constraint defined over variables $\bar{z}$. When variables are not important, we write $\varphi$.

**Translation into loss** Given a formula $\varphi$, we define the corresponding loss $\mathcal{L}(\varphi)$ recursively on the structure of $\varphi$. The obtained loss is non-negative: for any assignment $\bar{x}$ to the variables $\bar{z}$, we have $\mathcal{L}(\varphi)(\bar{x}) \in \mathbb{R}^{\geq 0}$. Further, the translation has two properties: (P1) any $\bar{x}$ for which the loss is zero ($\mathcal{L}(\varphi)(\bar{x}) = 0$) is a satisfying assignment to $\varphi$ (denoted by $\bar{x} \models \varphi$) and (P2) the loss is differentiable almost everywhere. This construction avoids pitfalls of other approaches (see Appendix B). We next formally define the translation rules. Formula $\varphi$ is parametrized by $\xi > 0$ which denotes tolerance for strict inequality constraints. Since comparisons are applied element-wise (i.e., on scalars), atomic constraints are transformed into a conjunction of scalar comparisons:

$$\mathcal{L}(t^1 \bowtie t^2) \quad := \quad \mathcal{L}\left(\bigwedge_{i=1}^{n} t_i^1 \bowtie t_i^2\right)$$

The comparisons $=$ and $\leq$ are translated based on a function $d \colon \mathbb{R} \times \mathbb{R} \to \mathbb{R}$ which is a continuous, differentiable almost everywhere, distance function with $d(x_1, x_2) \geq 0$ and $d(x_1, x_2) = 0 \leftrightarrow x_1 = x_2$:

$$\mathcal{L}(t^1 = t^2) \quad := \quad d(t^1, t^2); \qquad \mathcal{L}(t^1 \leq t^2) \quad := \quad \mathbf{1}_{t^1 > t^2} \cdot d(t^1, t^2)$$

Here, $\mathbf{1}_{t^1 > t^2}$ is an indicator function: it is 1 if $t^1 > t^2$, and 0 otherwise. The function $d$ is a parameter of the translation and in our implementation we use the absolute distance $|t^1 - t^2|$. For the other comparisons, we define the loss as follows: $\mathcal{L}(t^1 < t^2) = \mathcal{L}(t^1 + \xi \leq t^2)$, $\mathcal{L}(t^1 \neq t^2) = \mathcal{L}(t^1 < t^2 \vee t^2 < t^1)$, and $\mathcal{L}(t^1 > t^2)$ and $\mathcal{L}(t^1 \geq t^2)$ are defined analogously.

Conjunctions and disjunctions of formulas $\varphi$ and $\psi$ are translated into loss as follows:

$$\mathcal{L}(\varphi \wedge \psi) \quad := \quad \mathcal{L}(\varphi) + \mathcal{L}(\psi); \qquad \mathcal{L}(\varphi \vee \psi) \quad := \quad \mathcal{L}(\varphi) \cdot \mathcal{L}(\psi)$$

Note that $\mathcal{L}(\varphi \wedge \psi) = 0$ if and only if $\mathcal{L}(\varphi) = 0$ *and* $\mathcal{L}(\psi) = 0$, which by construction is true if $\varphi$ and $\psi$ are satisfied, and similarly $\mathcal{L}(\varphi \vee \psi) = 0$ if and only if $\mathcal{L}(\varphi) = 0$ *or* $\mathcal{L}(\psi) = 0$.

**Translating negations** Negations are handled by first eliminating them from the constraint through rewrite rules, and then computing the loss of their equivalent, negation-free constraint. Negations of atomic constraints are rewritten to an equivalent atomic constraint that has no negation (note that $\neq$ is not a negation). For example, the constraint $\neg(\boldsymbol{t}^1 \leq \boldsymbol{t}^2)$ is rewritten to $\boldsymbol{t}^2 < \boldsymbol{t}^1$, while negations of conjunctions and disjunctions are rewritten by repeatedly applying De Morgan's laws: $\neg(\varphi \wedge \psi)$ is equivalent to $\neg\varphi \vee \neg\psi$ and $\neg(\varphi \vee \psi)$ is equivalent to $\neg\varphi \wedge \neg\psi$.

With our construction, we get the following theorem:

**Theorem 1** *For all $\bar{x}$, if $\mathcal{L}(\varphi)(\bar{x}) = 0$, then $\bar{x}$ satisfies $\varphi$ ($\bar{x} \models \varphi$). Conversely, for any $\varphi$, there is a $\delta(\xi) \geq 0$ with $\lim_{\xi \to 0} \delta(\xi) = 0$ such that for all $\bar{x}$, $\mathcal{L}(\varphi)(\bar{x}) > \delta(\xi)$ implies that $\bar{x}$ does not satisfy $\varphi$ ($\bar{x} \not\models \varphi$).*

Essentially, as we make $\xi$ smaller (and $\delta$ approaches 0), we get closer to an if and only if theorem: if $\bar{x}$ makes the loss 0, then we have a satisfying assignment; otherwise, if $\bar{x}$ makes the loss $> \delta$, then $\bar{x}$ is not a satisfying assignment. We provide the proof of the theorem in Appendix A.

## 4 CONSTRAINED NEURAL NETWORKS

In this section, we present our method for training neural networks with constraints. We first define the problem, then provide our min-max formulation, and finally, discuss how we solve the problem. We write $[\varphi]$ to denote the indicator function that is 1 if the predicate holds and 0 otherwise.

**Training with constraints** To train with a single constraint, we consider the following maximization problem over neural network weights $\theta$:

$$\arg\max_{\theta} \mathbb{E}_{\boldsymbol{S}^1,\ldots,\boldsymbol{S}^m \sim \mathcal{D}} \left[ \forall \bar{z}.\ \varphi(\bar{z}, \bar{S}, \bar{c}, \theta) \right].$$

Here, $\boldsymbol{S}^1, \ldots, \boldsymbol{S}^m$ (abbreviated by $\bar{S}$) are independently drawn from an underlying distribution $\mathcal{D}$ and $\varphi$ is a constraint over variables $\bar{z}$, constants $\bar{S}$ and $\bar{c}$, and network weights $\theta$. This objective is bounded between 0 and 1, and attains 1 if and only if the probability the network satisfies the constraint $\varphi$ is 1. We extend this definition to multiple constraints, by forming a convex combination of their respective objectives: for $\boldsymbol{w}$ with $\sum_{i=1}^{t} w_i = 1$ and $w_i > 0$ for all $i$, we consider

$$\arg\max_{\theta} \sum_{i=1}^{t} w_i \cdot \mathbb{E}_{\boldsymbol{S}^1,\ldots,\boldsymbol{S}^m \sim \mathcal{D}} \left[ \forall \bar{z}.\ \varphi_i(\bar{z}, \bar{S}, \bar{c}, \theta) \right]. \tag{1}$$

As standard, we train with the empirical objective where instead of the (unknown) distribution $\mathcal{D}$, we use the training set $\mathcal{T}$ to draw samples.

To use our system, the user specifies the constraints $\varphi_1, \ldots, \varphi_t$ along with their weights $w_1, \ldots, w_t$. In the following, to simplify the presentation, we assume that there is only one constraint.

**Formulation as min-max optimization** We can rephrase training networks with constraints as minimizing the expectation of the *maximal violation*. The maximal violation is an assignment to the variables $\bar{z}$ which violates the constraint (if it exists). That is, it suffices to solve the problem $\arg\min_{\theta} \mathbb{E}_{\boldsymbol{S}^1,\ldots \boldsymbol{S}^m \sim \mathcal{T}} \left( \max_{\boldsymbol{z}^1,\ldots,\boldsymbol{z}^k} \neg\varphi(\bar{z}, \bar{S}, \bar{c}, \theta) \right)$. Assume that one can compute, for a given $\bar{S}$ and $\theta$, an optimal solution $\bar{x}^*_{\bar{S},\theta}$ for the inner maximization problem:

$$\bar{x}^*_{\bar{S},\theta} = \arg\max_{\boldsymbol{z}^1,\ldots,\boldsymbol{z}^k} [\neg\varphi(\bar{z}, \bar{S}, \bar{c}, \theta)]. \tag{2}$$

Then, we can rephrase the optimization problem in terms of $\bar{x}^*_{\bar{S},\theta}$:

$$\arg\min_{\theta} \mathbb{E}_{\boldsymbol{S}^1,\ldots \boldsymbol{S}^m \sim \mathcal{T}} [\neg\varphi(\bar{x}^*_{\bar{S},\theta}, \bar{S}, \bar{c}, \theta)]. \tag{3}$$

The advantage of this formulation is that it splits the problem into two sub-problems and the overall optimization can be seen as a game between an oracle (solving (2)) and an optimizer (solving (3)).

**Solving the optimization problems** We solve (2) and (3) by translating the logical constraints into differentiable loss (as shown in Sec. 3). Inspired by Theorem 1, for the oracle (Eq. (2)), we approximate the inner maximization by a *minimization* of the translated loss $\mathcal{L}(\neg\varphi)$:

$$\bar{x}^*_{\bar{S},\theta} = \arg\min_{\boldsymbol{z}^1,\ldots,\boldsymbol{z}^k} \mathcal{L}(\neg\varphi)(\bar{z}, \bar{S}, \bar{c}, \theta). \tag{4}$$

Given $\bar{x}_{\bar{S},\theta}$ from the oracle, we optimize the following loss using Adam (Kingma & Ba, 2014):

$$\mathbb{E}_{\boldsymbol{S}^1,\ldots,\boldsymbol{S}^m \sim \mathcal{T}} \left( \mathcal{L}(\varphi)(\bar{x}_{\bar{S},\theta}, \bar{S}, \bar{c}, \theta) \right). \tag{5}$$

**Constrained optimization** In general, the loss in (4) can sometimes be difficult to optimize. To illustrate, assume that the random samples are input-label pairs $(\boldsymbol{x}, y)$ and consider the constraint:

$$\varphi(\boldsymbol{z}, (\boldsymbol{x}, y), \theta) = ||\boldsymbol{x} - \boldsymbol{z}||_\infty \leq \epsilon \implies \mathrm{logit}^\theta(\boldsymbol{z})_y > \delta.$$

Our translation of this constraint to a differentiable loss produces

$$\mathcal{L}(\neg\varphi)(\boldsymbol{z}, (\boldsymbol{x}, y), \theta) = \max(0, ||\boldsymbol{x} - \boldsymbol{z}||_\infty - \epsilon) + \max(0, \mathrm{logit}^\theta(\boldsymbol{z})_y - \delta).$$

This function is difficult to minimize because the magnitude of the two terms is different. This causes first-order methods to optimize only a single term in an overly greedy manner, as reported by Carlini & Wagner (2017). However, some constraints have a closed-form analytical solution, e.g., the minimization of $\max(0, ||\boldsymbol{x} - \boldsymbol{z}||_\infty - \epsilon)$ can be solved by projecting into the $L_\infty$ ball. To leverage this, we identify logical constraints which restrict the variables $\boldsymbol{z}$ to convex sets that have an efficient algorithm for projection, e.g., line segments, $L_2$, $L_\infty$ or $L_1$ balls (Duchi et al., 2008). Note that in general, projection to a convex set is a difficult problem. We exclude such constraints from $\varphi$ and add them as constraints of the optimization. We thus rewrite (4) as:

$$\bar{x}_{\bar{S},\theta}^* = \underset{\boldsymbol{z}^1 \in D_1(\bar{S}),\ldots,\boldsymbol{z}^k \in D_k(\bar{S})}{\arg\min} \mathcal{L}(\neg\varphi)(\bar{z}, \bar{S}, \bar{c}, \theta), \tag{6}$$

where the $D_i$ denote functions which map random samples to a convex set. To solve (6), we employ Projected Gradient Descent (PGD) which was shown to have strong performance in the case of adversarial training with $L_\infty$ balls (Madry et al., 2017).

**Training procedure** Algorithm 1 shows our training procedure. We first form a mini-batch of random samples from the training set $\mathcal{T}$. Then, the oracle finds a solution for (4) using the formulation in (6). This solution is given to the optimizer, which solves (5). Note that if $\varphi$ has no variables ($k = 0$), the oracle becomes trivial and the loss is computed directly.

---

**Algorithm 1:** Training with constraints.

**input** : Training set $\mathcal{T}$, network parameters $\theta$, and a constraint $\varphi(\bar{z}, \bar{S}, \bar{c}, \theta)$

**for** *epoch = 1 to $n_{epochs}$* **do**

    Sample mini-batch of $m$-tuples
    $\bar{S} = \boldsymbol{S}^1, \ldots, \boldsymbol{S}^m \sim \mathcal{T}$.
    Using PGD, compute
    $\bar{x} \approx \arg\min_{\boldsymbol{z}^1 \in D_1(\bar{S}),\ldots,\boldsymbol{z}^k \in D_k(\bar{S})} \mathcal{L}(\neg\varphi)(\bar{z}, \bar{S}, \bar{c}, \theta)$.
    Perform Adam update with $\nabla_\theta \mathcal{L}(\varphi)(\bar{x}, \bar{S}, \bar{c}, \theta)$.

---

## 5 QUERYING NETWORKS

We build on DL2 and design a declarative language for querying networks. Interestingly, the hard-coded questions investigated by prior work can now be phrased as DL2 queries: neurons responsible for a prediction (Olah et al., 2018), inputs that differentiate networks (Pei et al., 2017), and adversarial examples (e.g., Szegedy et al. (2013)). We support the following class of queries:

$$\mathbf{find} \;\; \boldsymbol{z}^1(m^1), \ldots, \boldsymbol{z}^k(m^k) \;\; \mathbf{where} \;\; \varphi(\bar{z}) \;\; [\mathbf{init} \;\; \boldsymbol{z}^1 = \boldsymbol{c}^1, \ldots, \boldsymbol{z}^k = \boldsymbol{c}^k] \;\; [\mathbf{return} \;\; \boldsymbol{t}(\bar{z})]$$

Here, **find** defines the variables and their shape in parentheses, **where** defines the constraint (over the fragment described in Sec. 3), **init** defines initial values for (part or all of) the variables, and **return** defines a target term to compute at the end of search; if missing, $\boldsymbol{z}^1, \ldots, \boldsymbol{z}^k$ are returned. Networks (loaded so to be used in the queries) and constants are defined outside the queries. We note that the user can specify tensors in our language (we do not assume these are simplified to vectors). In queries, we write comma (**,**) for conjunction ($\wedge$); **in** for box-constraints and **class** for constraining the target label, which is interpreted as constraints over the labels' probabilities.

**Examples** Fig. 1 shows few interesting queries. The first two are defined over networks trained for CIFAR-10, while the last is for MNIST. The goal of the first query is to find an adversarial example `i` of shape $(32, 32, 3)$, classified as a truck (class 9) where the distance of `i` to a given deer image (`deer`) is between 6 and 24, with respect to the infinity norm. Fig. 1b is similar, but the goal is to find `i` classified as a deer where a specific neuron is deactivated. The last query's goal is to find `i` classified differently by two networks where part of `i` is fixed to pixels of the image `nine`.

```
find i[32, 32, 3]        find i[32, 32, 3]            find i[28, 28]
where i in [0, 255],     where i in [0, 255],         where i in [0, 1],
   class(NN(i)) = 9,        ‖i - deer‖∞ < 25,            i[0:9,:] = nine[0:9,:],
   ‖i - deer‖∞ < 25,       NN(i).l₁[0, 1, 1, 31] = 0,   class(NN1(i)) = 8,
   ‖i - deer‖∞ > 5         class(NN(i)) = 4             class(NN2(i)) = 9
```

   (a) Adversarial example.       (b) Neuron deactivated.       (c) Diffing networks.

Figure 1: DL2 queries enable to declaratively search for inputs satisfying constraints over networks.

**Solving queries** As with training, we compile the constraints to a loss, but unlike training, we optimize with L-BFGS-B. While training requires batches of inputs in PGD optimization, querying looks for *one* assignment, and thus there is more time to employ the more sophisticated, but slower, L-BFGS-B. We discuss further optimizations in Appendix C.

## 6 EXPERIMENTAL EVALUATION

We now present a thorough experimental evaluation on the effectiveness of DL2 for querying and training neural networks with logical constraints. Our system is implemented in PyTorch (Paszke et al., 2017) and evaluated on an Nvidia GTX 1080 Ti and Intel Core i7-7700K with 4.20 GHz.

### 6.1 TRAINING WITH DL2

We evaluated DL2 on various tasks (supervised, semi-supervised and unsupervised learning) across four datasets: MNIST, FASHION (Xiao et al., 2017), CIFAR-10, and CIFAR-100 (Krizhevsky & Hinton, 2009). In all experiments, one of the constraints was cross-entropy (see Sec. 4), to optimize for high prediction accuracy. For each experiment, we describe additional logical constraints.

**Supervised learning** We consider two types of constraints for supervised learning: *global constraints*, which have $z$-s, and *training set constraints*, where the only variables are from the training set (no $z$-s). Note that none of prior work applies to *global constraints* in general. Furthermore, because of limitations of their encoding explained in Sec. 2, they are not able to handle complex *training set constraints* considered in our experiments (e.g., constraints between probability distributions). To ease notation, we write random samples (the $S$-s) as $x^i$ and $y^i$ for inputs from the training set ($x^i$) and their corresponding label ($y^i$).

|  |  | MNIST | | FASHION | | CIFAR-10 | |
|---|---|---|---|---|---|---|---|
|  |  | Baseline | DL2 | Baseline | DL2 | Baseline | DL2 |
| Robustness$^{\text{T}}$ | P | 99.39 | 98.61 | 90.56 | 89.70 | 90.98 | 87.08 |
|  | C | 96.10 | 97.30 | 95.20 | 96.30 | 91.10 | 93.08 |
| Robustness$^{\text{G}}$ | P | 99.38 | 99.45 | 91.45 | 89.41 | 90.59 | 78.86 |
|  | C | 35.02 | 92.18 | 00.00 | 80.20 | 07.16 | 21.00 |
| Lipschitz$^{\text{T}}$ | P | 99.48 | 97.95 | 92.10 | 87.49 | 90.59 | 90.24 |
|  | C | 07.20 | 99.20 | 06.40 | 99.53 | 07.16 | 99.60 |
| Lipschitz$^{\text{G}}$ | P | 99.44 | 99.24 | 92.22 | 82.27 | 90.51 | 87.32 |
|  | C | 00.00 | 99.90 | 00.00 | 89.38 | 00.00 | 99.55 |
| C-similarity$^{\text{T}}$ | P | - | - | - | - | 91.52 | 90.70 |
|  | C | - | - | - | - | 89.10 | 99.60 |
| C-similarity$^{\text{G}}$ | P | - | - | - | - | 91.06 | 90.30 |
|  | C | - | - | - | - | 49.74 | 57.31 |
| Segment$^{\text{G}}$ | P | 98.16 | 97.44 | 88.13 | 87.18 | - | - |
|  | C | 18.97 | 37.70 | 21.80 | 48.18 | - | - |

Figure 2: Supervised learning, P/C is prediction/constraint accuracy.

For local robustness (Szegedy et al., 2013), the training set constraint says that if two inputs from the dataset are close (their distance is less than a given $\epsilon_1$, with respect to $L_2$ norm), then the KL divergence of their output probabilities is smaller than $\epsilon_2$:

$$\|x^1 - x^2\|_2 < \epsilon_1 \implies KL(p^\theta(x^1)\|p^\theta(x^2)) < \epsilon_2 \qquad \text{(Robustness}^{\text{T}}\text{)}$$

Second, the global constraint requires that for any input $x$, whose classification is $y$, inputs in its $\epsilon$ neighborhood which are valid images (pixels are between 0 and 1), have a high probability for $y$. For numerical stability, instead of the probability we check that the corresponding logit is larger than a given threshold $\delta$:

$$\forall z \in L_\infty(x, \epsilon) \cap [0, 1]^d.\ \text{logit}^\theta(z)_y > \delta \qquad \text{(Robustness}^{\text{G}}\text{)}$$

Similarly, we have two definitions for the Lipschitz condition. The training set constraint requires that for every two inputs from the training set, the distance between their output probabilities is less than the Lipschitz constant ($L$) times the distance between the inputs:

$$||p^\theta(\boldsymbol{x}^1) - p^\theta(\boldsymbol{x}^2)||_2 < L||\boldsymbol{x}^1 - \boldsymbol{x}^2||_2 \qquad \text{(Lipschitz}^\text{T}\text{)}$$

The global constraint poses the same constraint for valid images in the neighborhood $\boldsymbol{x}^1$ and $\boldsymbol{x}^2$:

$$\forall \boldsymbol{z}^1 \in L_\infty(\boldsymbol{x}^1, \epsilon) \cap [0,1]^d, \boldsymbol{z}^2 \in L_\infty(\boldsymbol{x}^2, \epsilon) \cap [0,1]^d.||p^\theta(\boldsymbol{z}^1) - p^\theta(\boldsymbol{z}^2)||_2 < L||\boldsymbol{z}^1 - \boldsymbol{z}^2||_2$$
$$\text{(Lipschitz}^\text{G}\text{)}$$

We also consider a training set constraint called *C-similarity*, which imposes domain knowledge constraints for CIFAR-10 networks. The constraint requires that inputs classified as a *car* have a higher probability for the label *truck* than the probability for *dog*:

$$y = \text{car} \implies p^\theta(\boldsymbol{x})_{\text{truck}} > p^\theta(\boldsymbol{x})_{\text{dog}} + \delta \qquad \text{(C-similarity}^\text{T}\text{)}$$

The global constraint is similar but applied for valid images in the $\epsilon$-neighborhood of $\boldsymbol{x}$:

$$\forall \boldsymbol{z} \in L_\infty(\boldsymbol{x}, \epsilon) \cap [0,1]^d.y = \text{car} \implies p^\theta(\boldsymbol{z})_{\text{truck}} > p^\theta(\boldsymbol{z})_{\text{dog}} + \delta \qquad \text{(C-similarity}^\text{G}\text{)}$$

Finally, we consider a *Segment* constraint which requires that if an input $\boldsymbol{z}$ is on the line between two inputs $\boldsymbol{x}^1$ and $\boldsymbol{x}^2$ in position $\lambda$, then its output probabilities are on position $\lambda$ on the line between the output probabilities:

$$\forall \boldsymbol{z}. \, \boldsymbol{z} = \lambda \cdot \boldsymbol{x}^1 + (1-\lambda) \cdot \boldsymbol{x}^2 \implies -\lambda \cdot \text{logit}^\theta(\boldsymbol{z})_{y^1} - (1-\lambda) \cdot \text{logit}^\theta(\boldsymbol{z})_{y^2} < \delta \quad \text{(Segment}^\text{G}\text{)}$$

Fig. 2 shows the prediction accuracy (P) and the constraint accuracy (C) when training with (i) crossed-entropy only (CE) and (ii) CE and the constraint. Results indicate that DL2 can significantly improve constraint accuracy (0% to 99% for Lipschitz$^\text{G}$), while prediction accuracy slightly decreases. The decrease is expected in light of a recent work (Tsipras et al. (2018)), which shows that adversarial robustness comes with decrease of prediction accuracy. Since adversarial robustness is a type of DL2 constraint, we suspect that we observe a similar phenomenon here.

**Semi-supervised learning** For semi-supervised learning, we focus on the CIFAR-100 dataset, and split the training set into labeled, unlabeled and validation set in ratio of 20/60/20. In the spirit of the experiments of Xu et al. (2018), we consider the constraint which requires that the probabilities of *groups of classes* have either very high probability or very low probability. A group consists of classes of a similar type (e.g., the classes *baby*, *boy*, *girl*, *man*, and *woman* are part of the *people* group), and the group's probability is the sum of its classes' probabilities. Formally, our constraint consists of 20 groups and its structure is:

| Method | Accuracy (%) |
|---|---|
| Baseline | 47.03 |
| Semantic loss | 51.82 |
| Rule distillation | 45.56 |
| DL2 | **53.48** |

Figure 3: Semi-supervised training.

$$(p_{people} < \epsilon \vee p_{people} > 1 - \epsilon) \wedge ... \wedge (p_{insects} < \epsilon \vee p_{insects} > 1 - \epsilon)$$

for a small $\epsilon$. We use this constraint to compare the performance of several approaches. For all approaches, we use the Wide Residual Network (Zagoruyko & Komodakis (2016)) as the network architecture. As a baseline, we train in a purely-supervised fashion, without using the unlabeled data. We also compare to *semantic loss* (Xu et al., 2018) and *rule distillation* (Hu et al., 2016). Note that this constraint is restricting the probability distribution and *not* samples drawn from it which makes other methods inapplicable (as shown in Sec. 2). As these methods cannot encode our constraint, we replace them with a closest approximation (e.g., the *exactly-one* constraint from Xu et al. (2018) for *semantic loss*). Details are shown in Appendix D. Fig. 3 shows the prediction accuracy on the test set for all approaches. Results indicate that our approach outperforms all existing works.

**Unsupervised learning** We also consider a regression task in an unsupervised setting, namely training MLP (Multilayer perceptron) to predict the minimum distance from a source to every node in an unweighted graph, $G = (V, E)$. One can notice that minimum distance is a function with certain properties (e.g., triangle inequality) which form a logical constraint listed below. Source is denoted as 0.

| Approach | MSE |
|---|---|
| *Supervised (regression)* | *0.0516* |
| Unsupervised (baseline) | 0.4938 |
| Unsupervised (with DL2) | **0.0998** |

Figure 4: Unsupervised training.

$$\forall v \in G, d(v) \geq 0 \wedge (\vee_{(v,v') \in E}(d(v) = d(v') + 1)) \wedge (\wedge_{(v,v') \in E}(d(v) \leqslant d(v') + 1))$$

| | MNIST | | | FASHION | | | CIFAR-10 | | | GTSRB | | | ImageNet | | |
|---|---|---|---|---|---|---|---|---|---|---|---|---|---|---|---|
| Nr. | #✓ | ⏱ | ⏱✓ | #✓ | ⏱ | ⏱✓ | #✓ | ⏱ | ⏱✓ | #✓ | ⏱ | ⏱✓ | #✓ | ⏱ | ⏱✓ |
| 1 | 10 | 0.4 | 0.4 | 10 | 0.4 | 0.4 | 10 | 0.4 | 0.4 | 10 | 0.6 | 0.6 | 10 | 1.6 | 1.6 |
| 2 | 10 | 1.0 | 1.0 | 10 | 1.00 | 1.00 | 10 | 1.2 | 1.2 | 10 | 3.0 | 3.0 | 10 | 80.7 | 80.7 |
| 3 | 10 | 1.0 | 1.0 | 10 | 0.9 | 0.9 | 10 | 3.4 | 3.4 | 10 | 3.1 | 3.1 | 10 | 81.0 | 81.0 |
| 4 | 10 | 1.5 | 1.5 | 10 | 1.6 | 1.6 | 9 | 9.8 | 1.9 | 0 | 120.0 | 0.0 | 10 | 94.4 | 94.4 |
| 5 | 10 | 1.0 | 1.0 | 10 | 1.0 | 1.0 | 8 | 16.5 | 1.1 | 10 | 3.2 | 3.2 | 10 | 80.2 | 80.2 |
| 6 | 9 | 15.7 | 4.1 | 8 | 25.5 | 1.9 | 9 | 5.7 | 1.7 | 8 | 27.4 | 4.2 | 9 | 81.4 | 77.2 |
| 7 | 10 | 1.0 | 1.0 | 10 | 1.0 | 1.0 | 10 | 1.1 | 1.1 | 10 | 3.0 | 3.0 | 10 | 74.0 | 74.0 |
| 8 | 6 | 48.6 | 1.0 | 6 | 51.7 | 6.2 | 9 | 5.9 | 1.0 | 9 | 15.8 | 4.2 | 10 | 78.4 | 78.4 |
| 9 | 10 | 1.4 | 1.4 | 10 | 1.5 | 1.5 | 8 | 10.5 | 1.7 | 0 | 120.0 | 0.0 | 10 | 86.6 | 86.6 |
| 10 | 10 | 1.8 | 1.8 | 10 | 1.7 | 1.7 | 10 | 2.6 | 2.6 | 0 | 120.0 | 0.0 | 10 | 92.2 | 92.2 |
| 11 | 6 | 50.0 | 3.3 | 7 | 42.0 | 8.7 | 7 | 35.0 | 17.1 | 0 | 120.0 | 0.0 | 8 | 96.6 | 90.7 |
| 12 | 10 | 2.0 | 2.0 | 10 | 2.0 | 2.0 | 9 | 28.8 | 18.7 | 10 | 6.2 | 6.2 | 0 | 120.0 | 0.0 |
| 13 | 5 | 63.5 | 7.1 | 7 | 39.3 | 4.7 | 7 | 30.2 | 7.6 | 9 | 21.9 | 11.0 | 0 | 120.0 | 0.0 |
| 14 | 0 | 120.0 | 0.0 | 0 | 120.0 | 0.0 | 7 | 68.21 | 46.01 | - | - | - | - | - | - |
| 15 | 3 | 79.08 | 71.27 | 9 | 35.02 | 25.58 | 7 | 66.02 | 42.88 | - | - | - | - | - | - |
| 16 | 1 | 108.2 | 2.0 | 1 | 108.2 | 2.0 | 8 | 34.4 | 13.1 | 4 | 75.6 | 9.0 | 0 | 120.0 | 0.0 |
| 17 | 10 | 2.9 | 2.9 | 10 | 3.2 | 3.2 | 5 | 61.6 | 4.0 | 0 | 120.0 | 0.0 | 0 | 120.0 | 0.0 |
| 18 | 10 | 4.0 | 4.0 | 10 | 4.0 | 4.0 | 7 | 50.6 | 24.9 | 0 | 120.0 | 0.0 | 0 | 120.0 | 0.0 |

Table 1: Results for queries: (#✓) number of completed instances (out of 10), ⏱ is the average running time in seconds, and ⏱✓ the average running time of successful runs (in seconds).

Additionally, we constrain $d(0) = 0$. Next, we train the model in an unsupervised fashion with the DL2 loss. In each experiment, we generate random graphs with 15 vertices and split the graphs into training (300), validation (150) and test set (150). As an unsupervised baseline, we consider a model which always predicts $d(v) = 1$. We also train a supervised model with the mean squared error (MSE) loss. Remarkably, our approach was able to obtain an error very close to supervised model, without using any labels at all. This confirms that loss generated by DL2 can be used to guide the network to satisfy even very complex constraints with many nested conjunctions and disjunctions.

## 6.2 QUERYING WITH DL2

We evaluated DL2 on the task of querying with constraints, implemented in TensorFlow. We considered five image datasets, and for each, we considered at least two classifiers; for some we also considered a generator and a discriminator (trained using GAN (Goodfellow et al., 2014a)). Table 3 (Appendix E) provides statistics on the networks. Our benchmark consists of 18 template queries (Appendix E), which are instantiated with the different networks, classes, and images. Table 1 shows the results (- denotes an inapplicable query). Queries ran with a timeout of 2 minutes. Results indicate that our system often finds solutions. It is unknown whether queries for which it did not find a solution even have a solution. We observe that the success of a query depends on the dataset. For example, queries 9-11 are successful for all datasets but GTSBR. This may be attributed to the robustness of GTSBR networks against the adversarial examples that these queries aim to find. Query 14, which leverages a discriminator to find adversarial examples, is only successful for the CIFAR dataset. A possible explanation can be that discriminators were trained against real images or images created by a generator, and thus the discriminator performs poorly in classifying arbitrary images. Query 15, which leverages the generators, succeeds in all tested datasets, but has only few successes in each. As for overall solving time, our results indicate that, successful executions terminate relatively quickly and that our system scales well to large networks (e.g., for ImageNet).

## 7 CONCLUSION

We presented DL2, a system for training and querying neural networks. DL2 supports an expressive logical fragment and provides translation rules into a differentiable (almost everywhere) loss, which is zero only for inputs satisfying the constraints. To make training tractable, we handle input constraints which capture convex sets through PGD. We also introduce a declarative language for querying networks which uses the logic and the translated loss. Experimental results indicate that DL2 is effective in both, training and querying neural networks.

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

# A    PROOF OF THEOREM 1

## A.1    $\mathcal{L}(\varphi)(\bar{x}) = 0$ IMPLIES SATISFACTION

We start by giving a proof for the if direction of the Theorem 1, i.e. if $\mathcal{L}(\varphi)(\bar{x}) = 0$, then $\bar{x}$ satisfies $\varphi$. The proof is by induction on the formula structure (we assume $\varphi$ is negation-free as negations can be eliminated as described in the text).

As a base case, we consider formulas consisting of a single atomic constraint.

- $\varphi = (t^1(\bar{x}) = t^2(\bar{x}))$
  If $d(t^1(\bar{x}), t^2(\bar{x})) = 0$, then by definition $t^1(\bar{x}) = t^2(\bar{x})$, and $\varphi$ is satisfied.
- $\varphi = (t^1(\bar{x}) \leq t^2(\bar{x}))$
  If $\mathbf{1}_{t^1(\bar{x}) > t^2(\bar{x})} \cdot d(t^1(\bar{x}), t^2(\bar{x})) = 0$, then by definition $t^1(\bar{x}) - t^2(\bar{x}) \leq 0$, and $\varphi$ is satisfied.
- $\varphi = (t^1(\bar{x}) < t^2(\bar{x}))$
  If $\mathbf{1}_{t^1(\bar{x}) + \xi > t^2(\bar{x})} \cdot d(t^1(\bar{x}) + \xi, t^2(\bar{x})) = 0$, then $t^1(\bar{x}) + \xi - t^2(\bar{x}) \leq 0$, and since $\xi > 0$, we get $t^1(\bar{x}) < t^2(\bar{x})$. Thus, $\varphi$ is satisfied.

As an induction step, we consider combination of formulas using single logical and or logical or operation.

- $\varphi \vee \psi$
  If $\mathcal{L}(\varphi) \cdot \mathcal{L}(\psi) = 0$, then either $\mathcal{L}(\varphi) = 0$ or $\mathcal{L}(\psi) = 0$. By the induction hypothesis, either $\varphi$ is satisfied or $\psi$ is satisfied, implying that $\varphi \vee \psi$ is satisfied.

- $\varphi \wedge \psi$
  If $\mathcal{L}(\varphi) + \mathcal{L}(\psi) = 0$, then (because $\mathcal{L}$ is non-negative) both $\mathcal{L}(\varphi) = 0$ and $\mathcal{L}(\psi) = 0$. By the induction hypothesis, $\varphi$ and $\psi$ are satisfied, implying that $\varphi \wedge \psi$ is satisfied.

## A.2    $\mathcal{L}(\varphi)(\bar{x}) \geq \delta(\xi)$ IMPLIES NON-SATISFACTION

As all variables come from a bounded set, it is easy to see that for every formula $\varphi$ there exists bound $T(\varphi)$ such that $\mathcal{L}(\varphi)(\bar{x}) \leq T(\varphi)$. In other words, loss can not be arbitrarily large for a fixed formula $\varphi$. Given formula $\varphi$, we will define $N(\varphi)$ such that the following statement holds:

**Lemma 1** *If $\mathcal{L}(\varphi)(\bar{x}) > N(\varphi) \cdot \xi$ then $\bar{x}$ does not satisfy $\varphi$ ($\bar{x} \not\models \varphi$).*

We prove Lemma 1 by induction on the formula structure. Base case of the induction is logical formula consisting of one atomic expression. In this case it is easy to see that if $\mathcal{L}(\varphi)(\bar{x}) > \xi$ then $\bar{x}$ does not satisfy the formula. This means we can set $N(\varphi) = 1$ for such formulas and statement of the theorem holds.

We distinguish between two cases:

- $\varphi = \varphi_1 \wedge \varphi_2$
  In this case we define:
  $$N(\varphi) = N(\varphi_1) + N(\varphi_2)$$

  Let $\bar{x}$ be an assignment which satisfies the formula $\varphi$. This implies that $\bar{x}$ satisfies both $\varphi_1$ and $\varphi_2$. From the assumption of induction we know that $\mathcal{L}(\varphi_1)(\bar{x}) < N(\varphi_1)\xi$ and $\mathcal{L}(\varphi_2)(\bar{x}) < N(\varphi_2)\xi$.
  Adding these inequalities (and using definitions of $\mathcal{L}$ and $N$) we get:
  $$\mathcal{L}(\varphi)(\bar{x}) = \mathcal{L}(\varphi_1)(\bar{x}) + \mathcal{L}(\varphi_2)(\bar{x}) < N(\varphi_1)\xi + N(\varphi_2)\xi = N(\varphi)\xi$$

- $\varphi = \varphi_1 \vee \varphi_2$
  In this case we define:
  $$N(\varphi) = \max\{N(\varphi_1)T(\varphi_2), N(\varphi_2)T(\varphi_1)\}$$

  Let $\bar{x}$ be an assignment which satisfies the formula $\varphi$. This implies that $\bar{x}$ satisfies one of $\varphi_1$ and $\varphi_2$. We can assume (without loss of generality) that $\bar{x}$ satisfies $\varphi_1$. From the

assumption of induction we know that $\mathcal{L}(\varphi_1)(\bar{x}) < N(\varphi_1)\xi$ and also $\mathcal{L}(\varphi_2)(\bar{x}) < T(\varphi_2)$. Multiplying these inequalities (and using definitions of $\mathcal{L}$ and $N$) we get:

$$\mathcal{L}(\varphi)(\bar{x}) = \mathcal{L}(\varphi_1)(\bar{x}) \cdot \mathcal{L}(\varphi_2)(\bar{x}) < N(\varphi_1)\xi \cdot T(\varphi_2) < N(\varphi)\xi$$

Thus, one can choose $\delta(\xi) = N(\varphi)\xi$. Then, $\lim_{\xi \to 0} \delta(\xi) = 0$ and for every assignment $\bar{x}$, $\mathcal{L}(\varphi)(\bar{x}) > \delta(\xi)$ implies that $\bar{x}$ does not satisfy $\varphi$ ($\bar{x} \not\models \varphi$), thus proving the Theorem 1.

To illustrate this construction we provide an example formula $\varphi = x^1 < 1 \wedge x^2 < 2$. The loss encoding for this formula is $\mathcal{L}(\varphi) = \max\{x^1 + \xi - 1, 0\} + \max\{x^2 + \xi - 2, 0\}$, where $\xi$ is the precision used for strong inequalities. For the given example our inductive proof gives $\delta(\xi) = 2\xi$. It is not difficult to show that assignments with loss greater than this value do not satisfy the formula. For example, consider $x^1 = 1 + \xi$ and $x^2 = 2 + 3\xi$. In this case $\mathcal{L}(\varphi)(\bar{x}) = 6\xi > \delta(\xi) = 2\xi$ and the assignment obviously does not satisfy $\varphi$. But also consider the assignment $x^1 = 1 - 0.5\xi$ and $x^2 = 2 - 0.5\xi$. In this case $\mathcal{L}(\varphi)(\bar{x}) > 0$ and $\mathcal{L}(\varphi)(\bar{x}) = \xi < \delta(\xi) = 2\xi$ and the assignment is indeed satisfying.

## B    COMPARISON OF DL2 WITH PRIOR APPROACHES

XSAT (Fu & Su, 2016) also translates logical constraints into numerical loss, but its atomic constraints are translated into non-differentiable loss, making the whole loss non-differentiable. Probabilistic soft logic (e.g., Cohen et al. (2017); Hu et al. (2016)) translates logical constraints into differentiable loss, which ranges between $[0, 1]$. However, using their loss to find satisfying assignments with gradient methods can be futile, as the gradient may be zero. To illustrate, consider the toy example of $\varphi(z) := (z = \left(\begin{smallmatrix} 1 \\ 1 \end{smallmatrix}\right))$. PSL translates this formula into the loss $\mathcal{L}_{\text{PSL}}(\varphi) = \max\{z_0 + z_1 - 1, 0\}$ (it assumes $z_0, z_1 \in [0, 1]$). Assuming optimization starts from $x = \left(\begin{smallmatrix} 0.2 \\ 0.2 \end{smallmatrix}\right)$ (or any pair of numbers such that $z_0 + z_1 - 1 \leq 0$), the gradient is $\nabla_z \mathcal{L}_{\text{PSL}}(\varphi)(x) = \left(\begin{smallmatrix} 0 \\ 0 \end{smallmatrix}\right)$, which means that the optimization cannot continue from this point, even though $x$ is not a satisfying assignment to $\varphi$. In contrast, with our translation, we obtain $\mathcal{L}(\varphi)(z) = |z_0 - 1| + |z_1 - 1|$, for which the gradient for the same $x$ is $\nabla_z \mathcal{L}(\varphi)(x) = \left(\begin{smallmatrix} -1 \\ -1 \end{smallmatrix}\right)$.

## C    OPTIMIZATION FOR QUERYING NETWORKS

Here we discuss how the loss compilation can be optimized for L-BFGS-B. While our translation is defined for arbitrary large constraints, in general, it is hard to optimize for a loss with many terms. Thus, we mitigate the size of the loss by extracting box constraints out of the expression. The loss is then compiled from remaining constraints. Extracted box constraints are passed to the L-BFGS-B solver which is then used to find the minimum of the loss. This "shifting" enables us to exclude a dominant part of $\varphi$ from the loss, thereby making our loss amenable to optimization. To illustrate the benefit, consider the query in Fig. 1a. Its box constraint, `i` **in** `[0,255]`, is a syntactic sugar to a conjunction with $2 \cdot 32 \cdot 32 \cdot 3 = 6,144$ atomic constraints (two for each variables, i.e., for every index $j$, we have $i_j \geq 0$ and $i_j \leq 255$). In contrast, the second constraint consists of 9 atomic constraints (one for each possible class different from 9, as we shortly explain), and the third and fourth constraints are already atomic. If we consider $6,155$ atomic constraints in the loss, finding a solution (with gradient descent) would be slow. For larger inputs (e.g., inputs for ImageNet, whose size is $224 \cdot 224 \cdot 3 > 150,000$), it may not terminate in a reasonable time. By excluding the box constraints from the loss, the obtained loss consists of only 11 terms, making it amenable for gradient optimization. We note that while a solution is not found (and given enough timeout), we restart L-BFGS-B and initialize the variables using MCMC sampling.

## D  EXPERIMENTS DETAILS

| | MNIST, FASHION | | | CIFAR-10 | | |
|---|---|---|---|---|---|---|
| | $\lambda$ | PGD Iterations | Params | $\lambda$ | PGD Iterations | Params |
| Robustness[T] | 0.2 | - | $\epsilon_1 = 7.8, \epsilon_2 = 2.9$ | 0.04 | - | $\epsilon_1 = 13.8, \epsilon_2 = 0.9$ |
| Robustness[G] | 0.2 | 50 | $\epsilon_1 = 0.3, \delta = 0.52$ | 0.1 | 7 | $\epsilon_1 = 0.03, \delta = 0.52$ |
| Lipschitz[T] | 0.1 | - | $L = 0.1$ | 0.1 | - | $L = 1.0$ |
| Lipschitz[G] | 0.2 | 50 | $L = 0.1$ | 0.1 | 5 | $L = 1.0$ |
| Classes[T] | - | - | - | 0.2 | - | $\delta = 0.01$ |
| Classes[G] | - | - | - | 0.2 | 10 | $\delta = 0.01$ |
| Segment[G] | 0.01 | 5 | $\epsilon = 100$ | - | - | - |

Table 2: Hyperparameters used for supervised learning experiment

Here we describe implementation details (including hyperaparameters) used during our experiments.

**Supervised learning**  For our experiments with supervised learning we used batch size 128, Adam optimizer with learning rate 0.0001. All other parameters are listed in 2. Additionally, for CIFAR-10 experiments we use data augmentation with random cropping and random horizontal flipping. Experiments with Segment constraints are done by first embedding images in 40-dimensional space using PCA. In lower dimensional space it is sensible to consider linear interpolation between images which is not the case otherwise. Note that this experiment is not performed for CIFAR-10 because we do not observe good prediction accuracy with baseline model using lower dimensional embeddings. This is likely because dimensionality of CIFAR-10 images is much higher than MNIST or FASHION.

We used ResNet-18 (He et al., 2016) for experiments on CIFAR-10 and convolutional neural network (CNN) with 6 convolutional and 2 linear layers for MNIST and FASHION (trained with batchnorm after each convolutional layer). The layer dimensions of CNN are (1, 32, 5x5) - (32, 32, 5x5) - (32, 64, 3x3) - (64, 64, 3x3) - (64, 128, 3x3) - (128, 128, 1x1) - 100 - 10 where (in, out, kernel-size) denotes a convolutional layer and a number denotes a linear layer with corresponding number of neurons.

**Semi-supervised learning**  All methods use the same Wide Residual Network model. We use depth 28 and widening factor 10. Neural network is optimized using Adam with learning rate 0.001. We use $\lambda = 0.6$ as weighting factor for DL2 loss.

For *semantic loss experiment* we follow the encoding from Xu et al. (2018). Please consult the original work to see how *exactly-one* constraint is encoded into semantic loss. Since *rule distillation* does not support our constraint, we use the following approximation (following notation from Hu et al. (2016)):

$$r_l(X, Y) = \sum_{Y' \in G(Y)} \sigma_\theta(Y') \tag{7}$$

We denote $G(Y)$ as set of labels sharing the same group as $Y$. Note that rule is meant to encourage putting more probability mass into the groups which already have high probability mass. This should result in the entire probability mass collapsed in one group in the end, as we want. We use $\pi_t = \max(0, 1.0 - 0.97^t)$ as mixing factor. Other constants used are $C = 1$ and $\lambda = 1$.

In this experiment, we used Wide Residual Networks (Zagoruyko & Komodakis, 2016) with $n$=28 and $k$=10 (i.e. 28 layers).

**Unsupervised learning**

Our model is the multilayer perceptron with $N*N$ input neurons, three hidden layers with 1000 neurons each and an output layer of N neurons. $N$ is the number of vertices in the graph, in our case 15. The input takes all vertices in the graph and the output is the distance for each node. The network uses ReLU activations and dropout of 0.3 after at each hidden layer. Network is optimized using Adam with learning rate 0.0001.

| Dataset | Type | Network | Architecture | Accuracy |
|---|---|---|---|---|
| **MNIST** | C | M_NN1: $[0,1]^{28\times28} \mapsto [0,1]^{10}$ | Tensorflow Tutorial | $0.992^\dagger$ |
| | C | M_NN2: $[0,1]^{28\times28} \mapsto [0,1]^{10}$ | M_NN1 with an additional layer | $0.990^\dagger$ |
| | G | M_G: $[-1,1]^{100} \mapsto [0,1]^{28\times28}$ | DC-GAN | - |
| | D | M_D: $[0,1]^{28\times28} \mapsto [0,1]$ | DC-GAN | - |
| | G | M_ACGAN_G: $[-1,1]^{100} \times \{0,\ldots,9\} \mapsto [0,1]^{28\times28}$ | AC-GAN | - |
| | D | M_ACGAN_D: $[0,1]^{28\times28} \mapsto [0,1] \times [0,1]^{10}$ | AC-GAN | - |
| **Fashion MNIST** | C | FM_NN1 : $[0,1]^{28\times28} \mapsto [0,1]^{10}$ | Tensorflow Tutorial | $0.917^\dagger$ |
| | C | FM_NN2 : $[0,1]^{28\times28} \mapsto [0,1]^{10}$ | FM_NN1 with an additional layer | $0.910^\dagger$ |
| | G | FM_G: $[-1,1]^{100} \mapsto [0,1]^{28\times28}$ | DC-GAN | - |
| | D | FM_D : $[0,1]^{28\times28} \mapsto [0,1]$ | DC-GAN | - |
| **CIFAR** | C | C_NN1 : $[0,255]^{32\times32\times3} \mapsto [0,1]^{10}$ | 4-layer-model | $0.712^\#$ |
| | C | C_NN2 : $[0,255]^{32\times32\times3} \mapsto [0,1]^{10}$ | 6-layer-model | $0.756^\#$ |
| | C | C_VGG : $[0,255]^{32\times32\times3} \mapsto [0,1]^{10}$ | VGG-16-based | $0.935^\#$ |
| | G | C_G : $[-1,1]^{100} \mapsto [0,255]^{32\times32\times3}$ | DC-GAN | - |
| | D | C_D : $[0,255]^{32\times32\times3} \mapsto [0,1]$ | DC-GAN | - |
| **GTSRB** | C | G_LeNet : $[0,1]^{32\times32} \mapsto [0,1]^{43}$ | based on LeNet | $0.914^\dagger$ |
| | C | G_VGG : $[0,1]^{32\times32} \mapsto [0,1]^{43}$ | based on VGG | $0.973^\dagger$ |
| **ImageNet** | C | I_V16 : $[0,255]^{224\times224\times3} \mapsto [0,1]^{1000}$ | VGG-16 from Keras | $0.715^*$ |
| | C | I_V19 : $[0,255]^{224\times224\times3} \mapsto [0,1]^{1000}$ | VGG-19 from Keras | $0.727^*$ |
| | C | I_R50 : $[0,255]^{224\times224\times3} \mapsto [0,1]^{1000}$ | ResNet-50 from Keras | $0.759^*$ |

Table 3: The datasets and networks used to evaluate DL2. The reported accuracy is top-1 accuracy and it was either computed by the authors ($*$), users that implemented the work ($\#$), or by us ($\dagger$). Note that for GTSRB the images have dimensions $32 \times 32 \times 3$, but the Cs take inputs of $32 \times 32 (\times 1)$, which are pre-processed grayscale versions.

# E ADDITIONAL DETAILS FOR SECTION 6.2

Here we provide statistics on the networks used in the experiments of Sec. 6.2, as well as the query templates.

**Dataset and networks** Our benchmark consists of five image datasets, each with different well-established neural networks' architectures. For each dataset, we consider at least two classifiers, and for some we also consider a generator and a discriminator (trained using GAN Goodfellow et al. (2014a)). We trained most networks ourselves, except for the C_VGG and the ImageNet classifiers, for which the weights were available to download. Table 3 summarizes the networks that we used, their architecture, and accuracy. Each row shows the dataset, the type of the network (classifier, generator, or discriminator), the network signature, and the architecture of the network. For example, the first row describes a classifier that takes as input images of size $28 \times 28$ pixels, each ranging between 0–1, and returns a probability distribution over ten classes.

| Query |
|---|
| 1  **eval** $N$(var) |
| 2  **find** i[shape]
   **where** $c$($N$(i))=c |
| 3  **find** i[shape]
   **where** $c$($N$(i))=c,
   pix_con |
| 4  **find** i[shape]
   **where** $c$($N$(i))=c,
   $N$(i).p[c] > 0.8,
   pix_con |
| 5  **find** i[shape]
   **where** $c$($N$(i))=c
   **init** i=var |
| 6  **find** i[shape]
   **where** $c$($N$(i))=c,
   pix_con,
   $\|i - var\|_\infty$ < dist
   **init** i=var |
| 7  **find** i[shape]
   **where** $c$($N$(i))=c,
   pix_con
   **init** i=var |
| 8  **find** i[shape]
   **where** $c$($N$(i))=c,
   i[mask] **in** range,
   i[nm]=var[nm]
   **init** i=var |
| 9  **find** i[shape]
   **where** $c$($N$(i))=c,
   pix_con,
   $N$(i).p[c] > 0.8
   **init** i=var |
| 10  **find** i[shape]
   **where** $c$($N$(i))=c,
   pix_con,
   $N$(i).p[c] > 0.8,
   $N$(i).p[$c_v$] < 0.1
   **init** i=var |
| 11  **find** i[shape]
   **where** $c$($N$(i))=c,
   pix_con,
   $N$(i).p[c] > 0.8,
   $N$(i).p[$c_v$] < 0.1,
   $\|i-var\|_\infty$ < dist
   **init** i=var |

| Query |
|---|
| 12  **find** i[shape]
   **where** pix_con,
   $c$($N_2$(i))=$c_2$,
   $c$($N_1$(i))=$c_1$ |
| 13  **find** i[shape]
   **where** pix_con,
   $c$($N_2$(i))=c,
   $\|i - var\|_\infty$ < dist,
   $c$($N_1$(i))=$c_v$,
   **init** i=var |
| 14  **find** i[shape]
   **where** $c$($N_1$(i))=$c_1$,
   $c$($N_2$(i))=$c_2$,
   $N_1$(i).p[$c_1$] > 0.5,
   $N_2$(i).p[$c_1$] < 0.1,
   $N_2$(i).p[$c_2$] > 0.5,
   $N_1$(i).p[$c_2$] < 0.1,
   pix_con,$D$(i) < 0.1 |
| 15  **find** i[100]
   **where** i **in** [-1,1],
   $c$($N_1$($G$(i)))=$c_1$,
   $N_1$($G$(i)).p[$c_1$]> 0.3,
   $c$($N_2$($G$(i)))=$c_2$,
   $N_2$($G$(i)).p[$c_2$]> 0.3 |
| 16  **find** i[shape]
   **where** $c$($N_1$(i))=$c_v$,
   $c$($N_2$(i))=c,
   i[mask] **in** range,
   i[nm]=var[nm]
   **init** i=var |
| 17  **find** i[shape]
   **where** $c$($N_1$(i))=$c_1$,
   $c$($N_2$(i))=$c_2$,
   $N_1$(i).p[$c_1$] > 0.5,
   $N_1$(i).p[$c_2$] < 0.1,
   pix_con |
| 18  **find** i[shape]
   **where** $c$($N_1$(i))=$c_1$,
   $c$($N_2$(i))=$c_2$,
   $N_1$(i).p[$c_1$] > 0.6,
   $N_1$(i).p[$c_2$] < 0.1,
   $N_2$(i).p[$c_2$] > 0.6,
   $N_2$(i).p[$c_1$] < 0.1,
   pix_con |

Figure 5: The template queries used to evaluate DL2.

# F  ADDITIONAL EXPERIMENTS

Here we provide further experiments to investigate scalability and run-time behavior of DL. For all experiments we use the same hyperparameters as in Section 6.2, but ran experiments I and II on a laptop CPU and experiment III on the same GPU setup as in Section 6.2 and increased the timeout to 300 s.

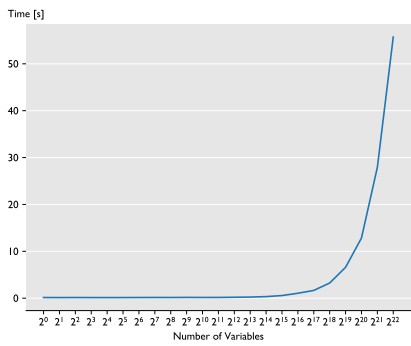
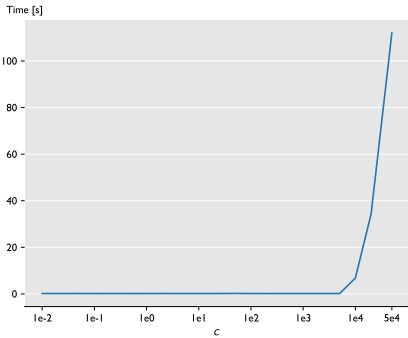

(a) Run-time for experiment 1. Runs up to $2^{13}$ variables are between $0.1 - 0.2$ s and don't show increase with number of variables. Afterwards growth is linear.

(b) Run-time for experiment 2. All runs up to $c = 5000$ take around $0.15$ s.

Figure 6: Experimental results for Experiments 1 and 2. Results are average over 10 runs with different random seed. All runs succeed.

**Experiment I: Number of variables** To study the run-time behavior in the number of variables we consider a simple toy query

```
find  i[c]
where  1000 < sum(i),  sum(i) < 1001
return  i
```

for different integers $c$. We execute this query for a wide range of $c$ values, 10 times each and report the average run-time in Figure 6a. All runs succeeded and found a correct solution. We observe constant run-time behavior for up to $2^{13}$ variables and linear run-time in the number of variables afterwards.

**Experiment II: Opposing constrains** To study the impact of (almost) opposing constraints we again consider a simple toy query

```
find  i[1]
where  i[0] < −c ∨ c < i[0]
return  i
```

for an integer $c$. This query requires optimizing two opposing terms until one of them is fulfilled. The larger $c$ the more opposed the two objectives are and indeed for $c \to \infty$ we would obtain an unsatisfiable objective. Again all runs succeeded and found a correct solution. In Figure 6b we present the average run-time over 10 runs for different $c$. Up to $c = 5000$ the run-time is constant with roughly $0.15$ s.

**Experiment III: Scaling in the number of constraints** To study the scaling of DL2 in the number of constraints consider the following query for an adversarial example:

For this experiment we consider the query:

```
FIND p[28, 28]
WHERE  class(M_NN1(clamp(p + M_nine, 0, 1))) = c
RETURN i, clamp(p + M_nine, 0, 1)
```

The query looks for an adversarial perturbation $p$ to a given image of a nine (`M_nine`) such that the resulting image gets classifies as class $c$. The query returns the found perturbation and the resulting image. The `clamp(I, a, b)` operation takes an input `I` and cuts off all it's values such that they are between $a$ and $b$.

Additionally we impose constraints the rows and columns of the image. For a row $i$ we want to enforce that the values of the perturbation vector are increasing from left to right:

| Row constraints | | | | | | | |
|---|---|---|---|---|---|---|---|
| k | 0 | 1 | 3 | 5 | 10 | 20 | 28 |
| ⏱✓ [s] | 1.01 | 2.63 | 5.77 | 9.63 | 19.77 | 44.61 | 84.49 |
| #✓ | 9 | 9 | 9 | 9 | 9 | 9 | 9 |
| **Column constraints** | | | | | | | |
| k | 0 | 1 | 3 | 5 | 10 | 20 | 28 |
| ⏱✓ [s] | 0.89 | 2.68 | 5.73 | 9.30 | 18.46 | 42.03 | 125.92 |
| #✓ | 9 | 9 | 9 | 9 | 9 | 9 | 9 |
| **Row & Column constraints** | | | | | | | |
| k | 0 | 1 | 3 | 5 | 10 | 20 | 28 |
| ⏱✓ [s] | 0.87 | 4.19 | 11.04 | 17.74 | 44.04 | 163.57 | 243.72 |
| #✓ | 9 | 9 | 9 | 9 | 9 | 9 | 4 |

Table 4: Run times for additional constraints on adversarial perturbation. #✓is the number of successful runs out of 9 and ⏱✓is the average run time over the successful runs in seconds. $k$ row or column constraints corresponds to 27 individual constraints in DL2 each. So the right most column adds 756 constraints for the first two settings and 1512 for the last.

```
p[i, 0] < p[i, 1], p[i, 1] < p[i, 2], p[i, 2] < p[i, 3], ...
```

For one row this yields 27 constraints. Further we consider a similar constraint for a column $j$, where we want the values to increase from to to bottom:

```
p[0, j] < p[1, j], p[1, j] < p[2, j], p[2, j] < p[3, j] ...
```

We apply these constraints on the first $k$ rows and columns of the image independently and jointly. For different $k$ we execute the query over all possible target classes $c \in \{0, \ldots, 8\}$ and report the average time in Table 4. The run-time is mostly linear for $k$ up to 20 but then jumps as we increase it to 28. The reason for this is likely that with many more constraints the solution spare grows sparer and sparser and it becomes hard to find an assignment. We observe that all queries, but for 5 in the case with $k = 28$ row and column constraints, could be solved. These 5 queries hit the 300 s timeout. Figure 7 shows a resulting image.

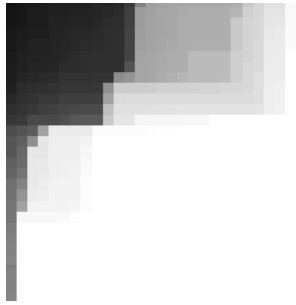

(a) The found perturbation $p$, scaled such that $-0.3$ corresponds to black and $0.3$ to white.

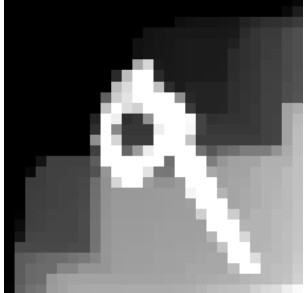

(b) The resulting image.

Figure 7: Found results for the full 28 row & column constraints and target class 6.

