# OpenReview forum: "DL2: Training and Querying Neural Networks with Logic"
_ICLR.cc/2019/Conference_

### Official Review · AnonReviewer1 · 2018-11-02

**Rating:** 6
**Confidence:** 2

**Review:**

In this paper the authors propose DL2 a system for training and querying neural networks with logical constraints

The proposed approach is intriguing but in my humble opinion the presentation of the paper could be improved. Indeed I think that the paper is bit too hard to follow.
The example at page 2 is not clearly explained.

In Equation 1 the relationship between constants S_i and the variables z is not clear. Is each S_i an assignment to z?

I do not understand the step from Eq. 4 to Eq. 6. Why does arg min become min?

At page 4 the authors state "we sometimes write a predicate \phi to denote its indicator function 1_\phi". I’m a bit confused here, when is the indicator function used in equations 1-6?

What kind of architecture is used for implementing DL2? Is a feedforward network used? How many layers does it have? How many neurons for each layer? No information about it is provided by authors.

It is not clear to me why DL2/training is implemented in PyTorch and DL2/querying in TensorFlow. Are those two separate systems? And why implementing them using different frameworks?

In conclusion, I’m a bit insecure about the rating to give to this paper, the system seems interesting, but several part are not clear to me.

[Minor comments]
It seems strange to me to use the notation L_inf instead of B_\epsilon to denote a ball.

In theorem 1. \delta is a constant, right? It seems strange to me to have a limit over a constant.

---

> ### Author Response · Authors · 2018-11-21
> **Clarification of Key Questions**
>
>
> → Presentation of the paper could be improved.
>
> A: We will update the paper this week, with better abstract / introduction and more clarification in the technical sections. In this we hope to put the work in context, convey the key ideas more clearly and provide additional explanations.
>
> → What is the differences between S^i and \bar{z}?
>
> A: In Equation 1 are S^i are drawn from the true data distribution and \bar{z} is a set of auxiliary variables. In general the values of z depend on the values of S (via the constraints). For example an auxiliary datapoint z^1 could be an adversarial example to the real data sample S^1. In the formulation in Example 1 this would mean that \phi for all possible assignments to z^1, while later we search for the assignments x that maximally violate \phi.
>
> → In the transition from Eq. 4 to Eq. 6., why does arg min become min?
>
> A: This is a mistake. Thank you for noticing it. We have updated min to arg min in Equation 6.
>
> → Why is the indicator function 1_{\phi} introduced on page 4?
>
> A: The indicator function mentioned on page 4 is indeed unused. In favor of readability we switched to iverson brackets [\phi], which evaluate to 1 if \phi is true and 0 else, in equations (1)- (3). We updated the paper correspondingly.
>
> →  Provide architecture details used in the evaluation.
>
> A: We updated our Appendix D with details on all models used for our experiments. DL2 proposes a loss function for training and querying any differentiable model with logical constraints. The number of neurons, layers etc. is the same as in the underlying model.
>
> → It is not clear to me why DL2/training is implemented in PyTorch and DL2/querying in TensorFlow. Are those two separate systems? And why implementing them using different frameworks?
>
> A: Our system provides both encoding of TensorFlow and PyTorch tensors to DL2 loss. We used TensorFlow for querying as its handling of compute graphs allows for a simpler implementation of the querying language. We will publish our framework with the entire system implemented in PyTorch.
>
> → In theorem 1. \delta is a constant, right? It seems strange to me to have a limit over a constant.
>
> A: We clarified this point in the write-up now: \delta is not a constant, but a function of \epsilon. Please take a look at the updated notation in our Theorem 1, which should make this explicit. We also provided a proof of Theorem 1 in the Appendix A which provides a constructive proof for the existence of \delta(\epsilon).

---

> > ### Author Response · Authors · 2018-11-23
> > **Update of PDF**
> >
> >
> > → Presentation of the paper could be improved.
> >
> > A: We now again updated the PDF with a new abstract and introduction which put the work in context and provide examples of DL2's benefits. We hope this improves the clarity of the examples and we are happy to take further suggestions into account.

---

### Official Review · AnonReviewer3 · 2018-11-02
**still needs improvement**

**Rating:** 5
**Confidence:** 4

**Review:**


The paper tackles the interesting problem of combining logical approaches with neural networks in the form of  translating a logical formula into a non-negative loss function for a neural network.
The approach is novel and more general than previous approaches and the math is sound. However, I feel that the method is not well presented. Sadly the introduction does not set the method into context or give a motivation. The abstract is very short and misses key information. Indeed, even the more technical parts sometimes lack clarity and assume familiarity with a wide range of methods.

The experiments are well thought out and show the promise of the method when encoding performance measures such as entropy into the constraints. It would have been interesting to additionally see other kinds of constraints such as purely logical formulas that do not have a specific aim (robustness or performance or otherwise) but simply state preconditions that should be fulfilled. It would furthermore be interesting to inspect the corner cases of the proposed method such as what happens if two constraints are nearly opposing each other and so on.



To conclude, the presented method is clearly novel and provides an interesting solution to a challenging problem. However the paper in the current form does not fully adhere to the standards of conferences such as ICLR. I suggest rewriting especially the abstract and the introduction and then submitting to a different venue as the approach itself seems promising. Additionally, as only very limited comparison experiments can be performed the method itself should be more thoroughly inspected by performing, for example, edge-case or time/number of constraints inspections.

Minor remarks:
Hyperparameters such as batch size not reported
Spelling mistake in line 2, page 2 “Lipschitz condition”
When mentioning “prior work” in the introduction a citation is needed.

---

> ### Author Response · Authors · 2018-11-21
> **Clarification of Key Questions**
>
>
> → Abstract, introduction and main body need  work to motivate the work and be more clear.
>
> A: We have already updated the abstract and introduction entirely to make it more understandable and to put the work in context. We also updated the main body of the paper. We will update the paper version this week.
>
> → What about opposing constraints and more experiments?
>
> A: We have investigated this in further experiments, found in Appendix F in the updated version. We investigated how DL2 runtime scales for a simple query with a different number of variables. To explore opposing constraints we optimize over the disjunction of directly opposing constraints with a single variable. By changing parameters we increase how far these two solutions are apart.
>
> Finally, to study the run-time behavior in the number of constraints we start from a query for an adversarial example and add up to 1512 additional constraints. We found that DL2 scales linear in most of these dimensions. For up to 8000 variables and most of the opposing constraints, all queries successfully finished in < 0.2s. We found that even adversarial examples with 1000 additional constraints still finish for all queries in < 160s. For 1500 additional constraints we had 4 of 9 queries complete successfully in about 240s each. The others hit the timeout of 300s.
>
> If the reviewer has further suggestions for experiments, we would be be happy to include these.

---

> > ### Author Response · Authors · 2018-11-23
> > **Update of PDF**
> >
> >
> > → Abstract, introduction and main body need  work to motivate the work and be more clear.
> >
> > A: We now again updated the PDF with a new abstract and introduction which put the work in context and provide examples of DL2's benefits. We believe we now motivate the approach and its strengths better and we are happy to take further suggestions into account.

---

### Official Review · AnonReviewer2 · 2018-11-04
**Interesting generalization of work on incorporating logical queries into neural networks, many compelling use cases**

**Rating:** 7
**Confidence:** 4

**Review:**

Summary
-------
This paper proposes DL2, a framework for turning queries over parameters and input, output pairs to neural networks into differentiable loss functions, and an associated declarative language for specifying these queries. The motivation for this work is twofold. The first is to allow for the specification of additional domain knowledge during training. For example, if a user expects that the predicted probabilities of some output classes should be correlated for all predictions, this constraint can be enforced during weight learning. Second, it allows users to search for specific inputs that satisfy specified conditions. In this way, DL2 can capture popular applications like searching for adversarial examples by querying for inputs close to a known input of class A but that the network predicts is class B with high confidence.

The paper provides a concise specification of the query language (a mixture of logical and numeric operators) and asserts a theorem that the given procedure for constructing the query loss produces a function such that anytime the function is 0, the constraints are satisfied. No proof is given, but I cannot see a counterexample. There is also a statement about the converse relationship, that when the loss is above some threshold it implies that the query is not satisfied.

Experiments are conducted on supervised, semi-supervised, and unsupervised computer vision tasks. I particularly liked the experiment on semi-supervised learning with CIFAR-100. By replacing labeled examples with domain knowledge about the relationships among classes in CIFAR-100, the paper demonstrates a compelling use case for DL2.

The primary technical challenge is the non-convex optimization required to search for a solution to a query. Experiments show that the loss functions created by DL2 are often solved quickly and correctly, but not always

Strengths
---------
The framework is expressive enough that many interesting use cases are clear, from specifying background knowledge during training to model inspection. The experiments cover a range of these use cases, demonstrating that the constructed optimization objectives usually work as intended.

Weaknesses
-----------
The statement in Theorem 1 regarding the converse case is unclear, because it says that the limit of \delta as \epsilon approaches zero is zero, but it is not explained what \epsilon is or how it changes. If \epsilon is the threshold that can often be used in the query, it is not obvious that every query contains exactly one \epsilon. If other cases exist, it is unclear how Theorem 1 applies.

It remains unknown how to handle the case when queries fail. AS the paper points out, if a query fails, it cannot be determined whether no solution exists or if the optimization simply failed to find a solution. Of course, this is a computationally hard in general.

Related Work
------------
There are a couple of points from related work that would be good to add to the paper.

First, the paper "Adversarial Sets for Regularising Neural Link Predictors" (Minervini et al., UAI17) is a prior paper that generates adversarial examples to handle restrictions on inputs which may not exist in the training set. The paper claims DL2 is the first to do this, but I believe this paper is an earlier example that does so, albeit for a particular problem. DL2 is certainly more general.

Second, the description of the limitations of rule distillation (Hu et al., ACL16), particularly in Appendix A is not fully accurate. The expressivity of PSL is greater than stated (see Bach et al., JMLR17 for a full description). In particular, the DL2 loss function for z = (1, 1) can be expressed exactly in PSL using what it calls arithmetic rules. It is not clear that this affects the findings of the semi-supervised learning experiment significantly, although I would appreciate a clarification of the authors. PSL by construction produces convex loss functions, and so the constraint that all outputs for a group of classes is either high OR low would probably not work well.

---

> ### Author Response · Authors · 2018-11-21
> **Clarification of Key Questions**
>
>
> → Clarification on Theorem 1 and \epsilon
>
> A: The \epsilon concerns only the *translation from logic to loss*. It denotes tolerance for strict inequality constraints. It is not part of the query and in practice is specified by the user once and for all. To avoid confusion with \epsilon that can appear in a query, we replaced \epsilon in the logic-to-loss translation with \xi. We updated the text and notation to clarify this point.
>
> → Proof of Theorem 1
>
> A: We provided a proof for both directions of Theorem 1. Our proof for the converse case is constructive and shows how to derive \delta for a fixed \epsilon. We also provide a small example to illustrate the relationship between \xi and \delta. All of this is found in Appendix A.
>
> → Expressivity of DL2 vs.  Hu et al., ACL16 vs. Bach et al. (JMLR17)
>
> A:  Thank you for the pointers to related work.
>
> The subset of PSL explicitly stated on page 3, equation 1 in Hu et al. (2016) describes Łukasiewicz Logic (LL) with an additional formula for conjunction. LL allows conjunction, disjunction and negation.
>
> Bach et al. (2017) extends LL, which we denote as PSL-HL. Specifically, it allows encoding of satisfaction of x^1 ≤ c by the minimization of max{x^1 − c, 0} (similar encodings for ≥ and = can be constructed). PSL-HL allows the conjunction of these arithmetic rules, but not their disjunction. Both LL and PSL-HL are described over domains of fuzzy booleans (x^i ∈ [0, 1]).
>
> DL2 is more general than PSL-HL:  (i) PSL-HL  is restricted to encoding of linear combinations of atoms (see Def. 15 in Bach et al. (2017)) and has closed-form solution, while DL2 is not as restrictive: it allows functions such as neural networks, constraining their outputs and relies on numerical optimization to find a solution, (ii) PSL-HL does not support disjunction over arithmetic rules while DL2 does, (iii) DL2 allows the domain of variables to be ℝ and not [0,1] as PSL-HL, (iv) more minor: PSL-HL considers a specific instantiation of d(t^1 , t^2), namely  |t^1 − t^2|,  while DL2 can consider other instantiations.
>
> Overall, while DL2 and PSL-HL permit similar encodings for some problems, DL2 is more general and more suitable to the domain of interacting with neural networks.
>
> LL indeed suffers from the problems outlined in Appendix B. Using PSL-HL arithmetic rules rather than LL in the particular example of Appendix B indeed produces the same encoding as DL2. Note that this is precisely because the example is over [0, 1] and does not contain disjunction.
>
> In the semi-supervised experiment on CIFAR-100 we used the LL variant of PSL. However, as the constraint is logical and not numerical, both LL and PSL-HL produce the same loss.
>
> An example of a query which cannot be encoded in PSL-HL can be found in our unsupervised learning experiment which contains disjunctions and uses R as a domain.
>
> References:
>
> [1] Bach, Stephen H., et al. "Hinge-loss markov random fields and probabilistic soft logic." arXiv preprint arXiv:1505.04406(2015).
>
> [2] Hu, Zhiting, et al. "Harnessing deep neural networks with logic rules." arXiv preprint arXiv:1603.06318 (2016).
>
> → DL2 and "Adversarial Sets for Regularising Neural Link Predictors" (Minervini et al., UAI17)
>
> A: Thank you for the reference. We make the claim that DL2, for the first time, queries for inputs outside the training set and uses them to globally train the network, in the context of [1] which introduces a logical loss that allows a user to encode a wide class of constraints, but not on examples outside the dataset. In this claim we did not consider adversarial training [2] which also produces new data outside the dataset (adversarial examples) and then trains over them. [3] proposes a similar min-max optimization, but further allows constraints as Horn clauses.
>
> We agree with the reviewer that DL2 is a generalization of these works which are restricted to their particular settings, but we will clarify the claim and amend the text to show the relation to these works.
>
> [1] Xu, Jingyi, et al. "A semantic loss function for deep learning with symbolic knowledge." arXiv preprint arXiv:1711.11157(2017).
>
> [2] Madry, Aleksander, et al. "Towards deep learning models resistant to adversarial attacks." arXiv preprint arXiv:1706.06083 (2017).
>
> [3] Minervini, Pasquale, et al. "Adversarial sets for regularising neural link predictors." arXiv preprint arXiv:1707.07596 (2017).
>
> → What happens if the query fails?
>
> A: If our solver fails to find a solution to a query, we cannot determine whether there is no solution or our approach failed to find one. To mitigate it, we run each query several times with different initialization points. In general, determining whether a formula over our fragment is satisfiable is not tractable as it is an instance of an SAT/SMT problem, with a very large number of variables and complex interactions (potentially multiple neural networks).

---

### Author Response · Authors · 2018-11-21
**Summary of Provided Clarifications**

We thank the reviewers for their insightful comments. Based on the reviews, we clarified the following key questions and updated the paper with a new revision:

- Added a proof of Theorem 1 in both directions, provided an example and updated notation to clarify what \delta is (Appendix A).
- Provided all architecture details used in our experiments (Appendix D).
- Performed additional experiments on scalability of DL2 as requested (Appendix F).
- Changed \epsilon to \xi in our notation to avoid confusion (Section 3).
- Fixed typos and minor notation issues pointed out by the reviewers.
- Clarified relation of DL2 to prior work, both logic (PSL) and training in our response to AnonReviewer2. If the reviewer is satisfied, we will update the paper with that.

Finally, we will provide a new introduction and abstract this week. We hope this can help put the work in context and make it more understandable. If the reviewers are satisfied with our answers, we will also update the paper to incorporate these as well. We are happy to answer more questions.

---

> ### Author Response · Authors · 2018-11-23
> **Update of PDF**
>
> We now again updated the PDF with a new abstract and introduction which put the work in context and provide examples of DL2's benefits.

---

### Meta-Review · Area_Chair1 · 2018-12-15
**A promising approach to include logical constraints in neural network training, but the writing is not quite ready yet.**

**Confidence:** 5
**Recommendation:** Reject

**Metareview:**

Unfortunately, this paper fell just below the bar for acceptance.  The reviewers all saw significant promise in this work, stating that it is intriguing, "novel and provides an interesting solution to a challenging problem" and that "many interesting use cases are clear".  AnonReviewer2 particularly argued for acceptance, arguing that the proposed approach provides a very flexible method for incorporating constraints in neural network training.  A concern of AnonReviewer2 was that there was no guarantee that this loss would be convex or converge to an optimum while statisfying the constraints.  The other two reviewers unfortunately felt that while the proposed approach was "interesting", "promising" and "intriguing", the quality of the paper, in terms of exposition, was too low to justify acceptance.  Arguably, it seems the writing doesn't do the idea justice in this case and the paper would ultimately be significantly more impactful if it was carefully rewritten.